# The Use of Virtual Reality to Reduce Stress among Inflammatory Bowel Disease Patients Treated with Vedolizumab

**DOI:** 10.3390/jcm10122709

**Published:** 2021-06-19

**Authors:** Konrad Lewandowski, Magdalena Kaniewska, Mariusz Rosołowski, Piotr Kucha, Grażyna Rydzewska

**Affiliations:** 1Clinical Department of Internal Medicine and Gastroenterology with Inflammatory Bowel Disease Unit, Central Clinical Hospital of the Ministry of the Inferior and Administration, 02-507 Warsaw, Poland; gastroenterologia@cskmswia.pl (K.L.); piotr.kucha@gmail.com (P.K.); grazyna.rydzewska@cskmswia.pl (G.R.); 2Department of Internal Medicine and Hypertension, Medical University of Bialystok, 15-276 Białystok, Poland; mariusz.rosolowski@umb.edu.pl; 3Department of Hypertension, Gastroenterology and Internal Medicine, Medical University of Bialystok Clinical Hospital, 15-276 Białystok, Poland; 4Collegium Medicum, Jan Kochanowski University, 25-317 Kielce, Poland

**Keywords:** virtual reality, biological treatment, vedolizumab

## Abstract

(1) Background: The use of virtual reality (VR) in improving patient comfort related to medical procedures in oncology patients raised the question of whether similar benefits could be obtained by patients with inflammatory bowel disease (IBD). (2) Methods: In this prospective, randomized, controlled, single-center clinical trial, a total of 90 patients with IBD treated with vedolizumab were enrolled and randomized in a 1:1 allocation to either the VR immersion group or the routine-treated group. The primary outcome was to evaluate whether VR could decrease stress and anxiety related to a medical procedure. The secondary outcome was to assess the safety of the VR. (3) Results: A statistically significant improvement in well-being and psychological comfort (*p* = 0.046), feeling of relaxation (*p* = 0.046), sense of influence on the treatment process (*p* < 0.001), improved perception of the way the drug works (*p* < 0.001), improved positive attitude while waiting for the next administration of the drug (*p* = 0.026), and increased motivation for treatment (*p* = 0.026) was noticed in the intervention group. There were no statistically significant differences in the incidence of complications in the intervention and control groups. (4) Conclusions: The use of VR had a positive effect on the reduction of stress associated with vedolizumab treatment and could improve compliance.

## 1. Introduction

Patients with inflammatory bowel disease (IBD), such as ulcerative colitis (UC) and Crohn’s disease (CD), are at a high risk of depression and anxiety [1,2,3]. Data on the frequency of these conditions are underestimated because of no routine screenings [1]. The incidence of depression and anxiety is higher in patients with IBD compared to the general population, and people with a long history of the disease are particularly at risk [4,5]. In the practice of treating patients with IBD, doctors mainly focus on treating the intestines [6,7]. Patients with IBD suffering from perianal fistulas are particularly vulnerable to depression [8]. Data from studies on a population of patients with chronic diseases and patients with IBD suggest that the presence of anxiety or depressive disorder significantly reduces treatment compliance [9,10]. Often, depression and anxiety may follow not only from the disease itself, but also from related medical procedures and treatment, especially when it requires additional hospital stays, with chemotherapy or biological treatment. Methods that can be used to treat depression and anxiety in patients with chronic diseases, including IBD, are: education of the physicians, pharmacological treatment, psychotherapy, exercise, and alternative methods [11,12] such as virtual reality (VR). VR is used in therapy to reduce the symptoms of depression, such as anxiety in chronic diseases [13]. The data show that VR helps to cope with anxiety, fear, and pain in some treatment regimens among children and adolescents and is considered by them as a very positive and interesting experience [14,15,16,17,18,19,20]. However, most studies focus on positive effects on managing treatment-related stress, increasing tolerance to medical procedures, and significantly reducing medication, distress, and fatigue following chemotherapy sessions [13,21,22,23,24].

The benefits of VR are particularly well documented in pain management [15]. An alternative mechanism to distraction of VR may be increasing the sense of control with the help of VR-assisted visualization. Research in this direction was carried out in the context of pain perception and its results may also bring benefits in the process of pharmacotherapy of IBD [24]. Guided imagery is a frequently used psychological technique in many clinical problems. These techniques are used in patients with chronic pain, as well as in cancer patients [25,26]. However, in several analyzed studies, an improvement in the emotional attitude of patients towards chemotherapy sessions was observed [16]. The techniques of directed imagery are also used in the psychological therapy of functional bowel disorders [27]. Existing studies indicate the possibility of using VR to reduce the subjective duration of chemotherapy sessions [21,27]. In one study, people undergoing chemotherapy rated its duration on average 34 min shorter than it actually was [13]. VR has been used to reduce stress in cancer patients and improve the quality of life [14]. Patients showed a lower level of anxiety and greater satisfaction with the radiotherapy sessions after the application of VR. Similar results for reducing anxiety have been observed after using VR distraction during chemotherapy in women with breast cancer [12].

Gut healing, which was used in this study, is designed to increase the subjective sense of control over the healing process, offering patients a virtual experience of attaching drug molecules to protein cells (so that protein cells do not pass into the digestive system or trigger an inflammatory process). The player actively supports this process in VR, which can also translate into a sense of control in reality. Like chemotherapy and other treatment methods, biological treatment seems to pose a lot of concern to IBD patients. We aimed to assess the effect of VR during biological therapy with vedolizumab, administered as an intravenous infusion, on the possible reduction of stress and anxiety associated with the procedure.

## 2. Materials and Methods

### 2.1. Study Design

This virtual reality immersion in patients with inflammatory bowel disease treated with vedolizumab intravenous infusion is a randomized, controlled, single-center clinical trial. A total of 90 patients, aged from 20 to 60 years, were randomized in a 1:1 allocation to either the VR immersion group (intervention group) or the routine-treated group (control group). Patients in the intervention group underwent a procedure of 15 min of VR once (gut healing), while patients in the control group were not subjected to the VR intervention during vedolizumab infusion.

The primary outcome of the study was to evaluate whether the VR intervention could decrease stress and anxiety related to medical procedure (vedolizumab infusion), measured with standardized questionnaires:Questionnaire for assessing patient’s attitude to drug administration sessions.Questionnaire to evaluate the application, allowing to assess how user-friendly the application is for patients.Questionnaire for measuring subjective psychological indicators. This will allow to measure the subjective effects of virtual reality application on the treatment process.Questionnaire to measure the sense of presence. The sense of presence is one of the constructs used to describe the subjective experience of being in some environment/space; the sense of presence is also independent of the place where a given individual is physically located. One can feel present in a given environment (e.g., a virtual forest), while physically that person is still in a hospital room. The sense of presence is treated as one of the best, subjective indicators of immersion in the virtual world. To measure the sense of presence in this study, items from the original English Igroup Presence Questionnaire (IPQ), translated into Polish, shortened, and adapted to the conditions of this study, were used.

The secondary outcome was to assess the safety of the VR intervention with a questionnaire on the occurrence of symptoms related to use of simulator:Questionnaire for measuring symptoms related to use of simulator.

### 2.2. Setting

The study was conducted at the Central Clinical Hospital of the Ministry of the Interior and Administration, a large comprehensive hospital in Warsaw, Poland. Each patient had a confirmed diagnosis of IBD and was treated according to the same standard of care by the same team of investigators.

### 2.3. Ethics

The study protocol was approved by the Bioethics Committee of the Central Clinical Hospital of the Ministry of the Interior and Administration in Warsaw, Poland. Anonymized data were analyzed.

### 2.4. Inclusion and Exclusion Criteria

To be eligible, the participants must have met all of the following criteria: (1) participants who had a confirmed diagnosis of inflammatory bowel disease, e.g., ulcerative colitis or Crohn’s disease, (2) were aged minimum 18 years, (3) and remained on vedolizumab therapy. The exclusion criteria were as follows: (1) participants who refused to sign the informed consent form, (2) participants with serious cardiopulmonary diseases, and (3) participants who had experienced physical discomfort, such as seizures, dizziness, eye twitching, and blackouts triggered by light flashes while wearing VR headset.

### 2.5. Randomization

After giving informed consent, the participants were randomized in a 1:1 allocation into the intervention group (immersion relaxation using gut healing game in VR) or the control group, based on a computer-generated random number concealed in a sealed envelope. One of the researchers conducted this process. The allocation was not be disclosed to other researchers who conducted the trial until the participant’s enrollment and assignment. In addition, the researchers who performed the statistical analyses were blinded to the group allocation.

### 2.6. Intervention Group

Participants were asked to wear VR headset (Samsung Economics Inc., Warsaw, Poland) and played the gut healing application for 15 min during vedolizumab infusion. Gut healing in VR combines the calm, relaxing properties of the game with the visualization of the biological treatment of IBD. While taking the drug in hospital setting, the patient can move to virtual reality with the help of goggles and a small controller. The scenery of the game is a virtual place reflecting the inside of the digestive system. The game is based on the mechanism of action of a humanized monoclonal IgG1 antibody (such as vedolizumab). Gameplay allows players to virtually experience blocking lymphocytes with a drug molecule. Just like the drug, the player also makes sure that lymphocytes do not enter the digestive system. Thanks to the virtual instruction, the player learns from the very beginning the basics of using the controller and the rules of the game. The following commands are displayed in stages with picture instructions. The player’s task in the virtual interior of the digestive system is to find and attract lymphocyte cells to himself/herself. To do this, the player aims a controller at them. The targeted cell is attracted to the player. The player then keeps the laser beam on the cell until it is close enough to administer the drug. The drug should be attached to all appropriate (green) receptors on the cell with the button on the controller. By arming the cell with the drug, the player prevents it from migrating to the intestines, which reflects the drug’s mechanism of action.

### 2.7. Control Group

Participants underwent vedolizumab infusion without VR immersion experience.

### 2.8. Statistical Analysis

Statistical analysis was conducted with the use of StatSoft version 13.0 (StatSoft Inc., Kraków, Poland). Nominal variables were presented as *n* (% frequency of group), while continuous variables as mean (SD) or median (Q1; Q3), depending on the normality of data distribution. Data normality was verified with the Shapiro–Wilk test and based on visual assessment of histograms. Groups were compared with the chi-square test for dichotomous variables and with the *t*-test or Mann–Whitney U test for continuous variables, as appropriate. All tests were two-sided, with α = 0.05.

## 3. Results

The distribution of sex, age, underlying disease (UC or CD), and previous use of biological treatment, in both the intervention and control groups, was similar (Table 1).

In the analysis of the questionnaire regarding patient’s attitude to drug administration we found a statistically significant improvement in well-being and psychological comfort (*p* = 0.046), feeling of relaxation (*p* = 0.046), a sense of influence on the treatment process (*p* < 0.001), improved perception of the way the drug works (*p* < 0.001), improved positive attitude while waiting for the next administration of the drug (*p* = 0.026), and increased motivation for treatment (*p* = 0.026). However, no statistical significance was found for the responses to questions regarding: feeling of prolonged infusion time (*p* = 0.103) or its shortening (*p* = 0.086), feeling of calmness (*p* = 0.128), feeling that a lot depends on the patient during treatment (*p* = 0.170), and feeling of calmness when approaching the next dose (*p* = 0.07) (Table 2).

The study group also completed the application evaluation questionnaire, which used the following scale: minimum score: 1—strongly disagree, and maximum score: 5—strongly agree. The first question concerned the length of time needed to master the rules of the game, which was rated at 3.82. Then, the degree of difficulty was assessed as “easy to use”, with the mean score of 3.80, and as too difficult, with the mean score being 2.22. The mean score for “the game level was well matched” was 3.71 and for “it was too easy” it was 3.58. Then, subjective feelings of patients about using the application were assessed. The questions concerned the following issues: feeling of pleasure (3.96), feeling of comfort (3.96), level of relaxation (3.96), and whether it was entertaining (3.80) (Table 3).

In addition, a question was asked about the willingness to use the application again. Here, 62.2% of respondents said yes, while 37.8% answered that they are not sure. Surprisingly, no patient in the study answered no (Table 4).

In order to assess the patients’ virtual reality, a questionnaire measuring the sense of presence was conducted. The answers to the questions asked were scored 1—I strongly disagree, to 5—I strongly agree. As regards questions in Table 5, the following answers were given: 1—3.56; 2—3.56; 3—2.40; 3—2.44; 4—3.58; 5—3.56; 6—3.91; 7—3.82; 8—3.38 and 9—3.60 (Table 5).

Another questionnaire concerned the safety of patients using VR through the application, i.e., the occurrence of symptoms related to use of simulator. Potential symptoms that could occur after the use of VR were listed, and patients were asked to grade them on a 4-point scale: 1—not at all, 2—mild, 3—moderate, and 4—significant. No statistical significance was found for the occurrence of the symptoms related to use simulator (general discomfort, fatigue, headache, eye strain, difficulty focusing, increased salivation, sweating, nausea, difficulty concentrating, feeling of “fullness of the head”, blurred vision, dizziness with the eyes closed, dizziness with the eyes open, giddiness, and stomach awareness) in both groups (Table 6).

## 4. Discussion

Findings from our study suggest a positive effect of VR during vedolizumab infusion. Detected statistical significance regarding the improvement of well-being and psychological comfort, the feeling of relaxation, the sense of influence on the treatment process, improved understanding of the drug effect, and improved positive attitude while waiting for subsequent drug administration clearly show a reduction in anxiety and stress related to treatment and medical procedures. Additionally, the fact that statistical significance was also noted in increasing the patient’s motivation for treatment may significantly improve the compliance, which, as we all know, positively influences the therapeutic process (Table 1).

According to the questionnaire evaluating the application, the system was easy to use, and the game was even rated as too simple. The questionnaire also assessed the subjective feelings of patients on the game as pleasant, feeling comfortable and relaxed, with a significant number of the respondents being interested in the game. Interestingly, in the control group, a lack of positive attitude and motivation to continue the treatment was noticed. This may be related to higher levels of stress and anxiety related not only to the treatment itself, but may be due to the fact of being in hospital, waiting for a medical procedure (unfavorable blood collection, drug infusion and its duration, appearance of possible side effects) (Table 2).

A vast majority of patients, i.e., 62.2%, expressed their will to play the game again, which additionally confirms the above conclusions. Only 37.8% answered that they were not sure if they would like to use VR again. At the same time, none of the respondents ruled out re-use of the application.

It was found that the application gave patients a sense of being transferred to virtual reality. The visual experience was assessed as real, the patients did not have an impression of looking at pictures, but still had a sense of real reality (Table 3 and Table 4).

Another important aspect was safety assessment in terms of the occurrence of symptoms related to use of simulator. There were no statistical differences between the study group and the control group (Table 5).

In 2003, Schneider et al. studied the effectiveness of VR in a group of older women with breast cancer, as a distraction during chemotherapy infusion. The study group was smaller than ours and included 16 patients. They were much older than our patients, i.e., 57.7 vs. 34.7 years, respectively. It was assumed that VR may be a potential factor that distracts patients from chemotherapy infusions and thus reduces the stress associated with the medical procedure. Two days after being in VR, a survey was conducted in which questions were asked about its effectiveness in distracting from the infusion and in reducing the sense of stress, and whether it has a lasting effect. Patients rated VR as a method that effectively diverted attention from chemotherapy with statistical significance. Moreover, there was a statistically significant reduction in the symptoms of distress, fatigue, and anxiety, which were measured using standardized questionnaires. No statistical significance was found for the answer to the second question. Similar results regarding the effectiveness of VR were obtained in our study. A particular improvement in the study by Schneider et al. was found for Symptom Distress Scale (SDS), State-Anxiety Inventory for Adults (SAI), and Revised Piper Fatigue Scale (PFS) scores [13]. In our study, statistical significance was found for the responses to questions regarding the improvement of well-being and psychological comfort, the feeling of relaxation, and improved positive attitude while waiting for the next administration of the drug, which clearly indicates the effectiveness of VR in reducing negative emotions associated with taking infusions of drugs.

Espinoza et al. conducted a study on 33 hospitalized patients to assess the effects of VR on symptoms of depression and anxiety, as well as levels of joy and relaxation. The intervention with VR lasted four sessions of 30 min over one week. During these sessions, two virtual environments were used to induce joy or relaxation. Symptoms of depression and anxiety (Hospital Anxiety and Depression Scale, HADS) and the level of happiness (Fordyce Scale) were assessed before and after the VR intervention. Visual analog scales (VAS) were also used to assess the emotional state and physical discomfort before and after each session. After the VR intervention, there was a significant improvement in distress and happiness levels. An increase in positive emotions and a decrease in negative emotions after the sessions were also detected. There was a significant reduction in the level of anxiety and depression (depression scale, *t* = 2747; *p* = 0.012 and total HADS, *t* = 2440; *p* = 0.024) and a significant increase in the level of happiness after the intervention (happiness intensity, *t* = −2.116; *p* = 0.047 and total happiness, *t* = −2.055; *p* = 0.05). In our study, we found similar results, i.e., improvements in well-being and psychological comfort (*p* = 0.046), as well as feeling of relaxation (*p* = 0.046). In a study by Espinoza et al., the sense of influence on the treatment process, the perception of the effect mechanism of the drug, attitude towards waiting for the next administration of the drug and motivation for treatment have not been studied. They only focused on the emotional state of feeling happy or sad. The parameters we examined were, however, important, especially in terms of compliance with the patient, which is a very important component of the therapeutic process. Their study also highlighted the difference in terms of the patient’s stay in the hospital. Our patients were treated on a one-day basis, while in the study by Espinoza et al. it was a one-day hospitalization [14].

Marquess et al. showed that patient education before treatment reduces anxiety and improves understanding. They conducted a pilot study that assessed the impact of VERTTM, an educational tool in a virtual environment, on these endpoints. Twenty-two patients with prostate cancer treated with radiotherapy were enrolled in the study and completed a 16-question questionnaire on understanding/anxiety. Patients switched to VERTTM, modeled on a flight simulator, using realistic sounds and views. The conclusions found a reduction in anxiety and a better understanding of treatment, which affect patient satisfaction and may contribute to improving the quality of health services. The implementation of virtual simulation effectively improved the understanding of treatment and reduced anxiety. Many studies to date have shown that patient education about the diagnostic and therapeutic process is essential to reduce the sense of anxiety. In view of the similar results compared to our study, where we found improved perception of the way the drug works (*p* < 0.001), improved positive attitude while waiting for the next administration of the drug (*p* = 0.026), and increased motivation for treatment (*p* = 0.026), it seems that there is a need for an interactive, patient-friendly solution that could be used in everyday practice [28].

Limitations of this study must be taken into account. First, this was a single center, randomised, but not blinded study. Second, the results of the study could not be easily extrapolated on the entire population of patients treated with biologics, since the only device that we used was dedicated to vedolizumab.

## 5. Conclusions

The analysis of the above data supports the initial hypothesis, and the implementation of virtual reality intervention during vedolizumab infusion significantly increased understanding of the therapy process and reduced stress in the treated group. A decrease in patient anxiety towards infusions could improve medication compliance and decrease disease burden while providing a decrease in the cost of healthcare.

## Figures and Tables

**Table 1 jcm-10-02709-t001:** Characteristics of the participants in the intervention and control group.

Variable	All (*n* = 90)	Control (*n* = 45)	Intervention (*n* = 45)	*p*-Value
Sex, female, n (%)	36 (40)	15 (33.3)	21 (46.7)	0.197
Age, y, mean (SD)	34.7 (9.9)	34 (9.8)	35.4 (10.9)	0.499
Diagnosis				0.506
Ulcerative colitis	59 (65.6)	31 (68.9)	28 (62.2)	
Crohn’s disease	31 (34.4)	14 (31.1)	17 (37.8)	
Biological treatment experienced	62 (68.9)	33 (73.3)	29 (64.4)	0.362

**Table 2 jcm-10-02709-t002:** The subjective psychological indicators questionnaire summary.

Variable (1-I Totally Disagree, 5-I Totally Agree)	Rank Sum Intervention	Rank Sum Control	*p*-Value
It seemed to me that the drug injection was too long	1,845,000	2,250,000	0.103
It seemed to me that the drug injection passed quickly	2,260,500	1,834,500	0.086
I felt calm during drug injection	2,236,500	1,858,500	0.128
I felt like a lot of things were up to me	2,218,000	1,877,000	0.170
I felt physically well and comfortable	2,295,000	1,800,000	0.046
I felt relaxed	2,295,000	1,800,000	0.046
I thought I was able to control the treatment process	2,506,000	1,589,000	<0.001
During the injection I was imaging how the drug works inside my body	2,466,000	1,629,000	<0.001
I’m expecting the next drug injection with a positive attitude	2,324,500	1,770,500	0.026
I’m expecting the next drug injection being strongly motivated to continue the treatment	2,324,500	1,770,500	0.026
I’m calm about the next drug injection	2,272,500	1,822,500	0.07

**Table 3 jcm-10-02709-t003:** The game evaluation questionnaire summary.

Variable (1-I Totally Agree, 5-I Totally Disagree)	Mean Score
I needed a lot of time to learn how to play	3.82
The application was easy to use	3.80
The game was too difficult	2.22
The difficulty level was appropriate	3.71
The game was too easy	3.58
Playing the game was enjoyable	3.96
I felt comfortable when playing the game	3.96
I think the game can help to relax	3.96
I think the game is entertaining	3.80

**Table 4 jcm-10-02709-t004:** The game evaluation questionnaire summary.

Would You Like to Use the Application Again?	No. of Patients
Yes	28 (62.2%)
Don’t know	17 (37.8%)
No	0 (0.00%)

**Table 5 jcm-10-02709-t005:** The presence questionnaire summary.

Variable (1-I Totally Disagree, 5-I Totally Agree)	Mean Score
It seemed to me like I was inside a virtual reality	3.56
It seemed to me like a virtual world surrounded me	3.56
Playing the game was like viewing photos	2.40
I didn’t feel like I was in a virtual space	2.44
It seemed to me that I fully control the particles not just navigating them from outside	3.58
I felt present in a virtual reality	3.56
I was totally aware of the real world around me	3.91
I was constantly paying attention to the real world around me	3.82
I got completely involved with the virtual world	3.38
The virtual reality seemed real to me	3.60

**Table 6 jcm-10-02709-t006:** The symptoms related to use of simulator questionnaire summary.

Variable (1-None, 2-Slight, 3-Moderate, 4-Severe)	Rank Sum Intervention	Rank Sum Control	*p*-Value
General discomfort	2,072,500	2,022,500	0.84
Fatigue	2,072,500	2,022,500	0.84
Headache	2,139,500	1,955,500	0.46
Eye strain	2,011,500	2,083,500	0.77
Difficulty focusing	2,015,000	2,080,000	0.79
Increased salivation	2,047,500	2,047,500	0.99
Sweating	2,070,000	2,025,000	0.86
Nausea	2,115,000	1,980,000	0.59
Difficulty concentrating	1,999,500	2,095,500	0.70
“Fullness of the head”	1,999,500	2,095,500	0.70
Blurred vision	2,025,000	2,070,000	0.86
Dizzy (eyes open)	2,047,500	2,047,500	0.99
Dizzy (eyes closed)	2,047,500	2,047,500	0.99
Giddiness	2,047,500	2,047,500	0.99
Stomach awareness	2,047,500	2,047,500	0.99

## Data Availability

The data presented in this study are available on request from Magdalena Kaniewska. The data are not publicly available due to privacy restrictions.

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
