# Peer review of "The Use of Virtual Reality to Reduce Stress among Inflammatory Bowel Disease Patients Treated with Vedolizumab"

_jcm, 2021, doi:10.3390/jcm10122709_

Round 1

Reviewer 1 Report

  • “Patients with IBD suffering from perivascular fistulas”

perivascular?!

  • Provide the age range of the included population

  • Why did you find a difference between “calm” and “relaxed”?

  • Why, regarding a treatment with a good safety profile and good efficacy, like vedolizumab, your group of control have so poor scores about “I’m expecting the next drug injection with a positive attitude” and “I’m expecting the next drug injection being strongly motivated to continue the treatment”

  • How do you explain that the score for “I needed a lot of time to learn how to play” is very similar to the score for “The application was easy to use”

  • “Only 7.8% answered that they were not 255 sure if they would like to use VR again.”

37.8% !!

  • Add the criticisms to your study

Author Response

Review 1

Thank You very much for Your review. Below we refer to every comments:

  1. Patients with IBD suffering from perivascular fistulas” perivascular??

Of course this was corrected (line 36). Sorry for the mistake

  1. Provide the age range of the included population.

The age range of the study population was from 20 to 60 years, this is updated on line 82.

  1. Why did you find a difference between “calm” and “relaxed”?

The questionnaires were designed by the team of psychologists. They were to serve as a standardized methods of emotional evaluation related to the use of virtual reality. The word calm defined as not showing or feeling nervousness, anger, or other strong emotions, and relaxed as free from tension and anxiety.

As You can see in the Table 2, the answers were similar for both.

  1. Why, regarding a treatment with a good safety profile and good efficacy, like vedolizumab, your group of control have so poor scores about “I’m expecting the next drug injection with a positive attitude” and “I’m expecting the next drug injection being strongly motivated to continue the treatment”.

The fact that the control group scored poorly on "I'm expecting the next drug injection with a positive attitude" and "I'm expecting the next drug injection being strongly motivated to continue the treatment" may be related to higher levels of stress and anxiety related not only to the treatment itself, it may be due to the fact of being in hospital, waiting for a medical procedure (unfavorable blood collection, drug infusion and its duration, appearance of possible side effects). The presence of stress and anxiety is presented in the introduction to this paper. In addition, the results themselves may support our hypothesis. The control group also had low scores on: "It seemed to me that the drug injection passed quickly", "I felt calm during drug injection", "I felt like a lot of things were up to me", "I felt physically well and comfortable ", "I felt relaxed" and "I thought I was able to control the treatment process". The explenation was included in the discussion, lines 231 - 236. .

  1. How do you explain that the score for “I needed a lot of time to learn how to play” is very similar to the score for “The application was easy to use”?

 "I needed a lot of time to learn how to play" was defined as the time needed to learn how the equipment works, how the game works and how it works. Patients rated it 3.82, which means that it took some time. Then we asked patients about "The application was easy to use", i.e. whether playing in the application caused a problem after learning the above knowledge. Patients rated the game fairly easy at 3.80 points. As You can see in the article, the answers were similar for both (Table 3).

  1. “Only 7.8% answered that they were not 255 sure if they would like to use VR again.” 37.8%!! Of course this was corrected (line 239). Sorry for the mistake.

  1. Add the criticisms to your study.

Updated on 312-315 lines. Limitations of this study must be taken into account. First, that was a single center, randomised, but not blinded study. Second, the results of the study could not be easily extrapolated on the entire population of patients treated with biologics, since the only device, that we used was dedicated to vedolizumab.

Reviewer 2 Report

The authors have conducted a very interesting study which involves some cutting-edge technology and its implementation in IBD care.

I do recommend the authors review the manuscript in detail for English comprehension related errors and correct them, some of them are listed in the following lines: 13, 17, 18, 26, 46, 52, 57, 63-65, 185, 240, 262 (please rephrase simulator disease symptoms to symptoms related to use of simulator).

Line 185, please change to: The distribution of sex, age, underlying disease (UC or CD) and previous use of biological treatment; in the treatment and control groups was similar.

Please reduce the size of the introduction to 2-3 paragraphs.

The authors need to add in conclusions that decrease in patient anxiety towards infusions will improve medication compliance and decrease disease burden along with a decrease in cost towards healthcare.

Author Response

Review 2:

Thank You very much for Your review. Below we refer to every comments:

  1. I do recommend the authors review the manuscript in detail for English comprehension related errors and correct them, some of them are listed in the following lines: 13, 17, 18, 26, 46, 52, 57, 63-65, 185, 240, 262 (please rephrase simulator disease symptoms to symptoms related to use of simulator).

The originally submitted manuscript was revised by an English native-speaker. In the updated version, "simulator disease symptoms" has been replaced with "symptoms related to use of simulator" (lines 105, 106, 209 and 243).

  1. Line 185, please change to: The distribution of sex, age, underlying disease (UC or CD) and previous use of biological treatment; in the treatment and control groups was similar.

Updated as recommended (line 167).

  1. Please reduce the size of the introduction to 2-3 paragraphs. The introduction was evidently too long, it was significantly shortened from 865 words to 609 and currently contains 3 paragraphs.

Updated as recommended

  1. The authors need to add in conclusions that decrease in patient anxiety towards infusions will improve medication compliance and decrease disease burden along with a decrease in cost towards healthcare.

Updated as recommended (lines 317 – 322).

Round 2

Reviewer 1 Report

Thank you for the corrections.